# The Role of Connexin 43 in Renal Disease: Insights from In Vivo Models of Experimental Nephropathy

**DOI:** 10.3390/ijms232113090

**Published:** 2022-10-28

**Authors:** Elena Roger, Louis Boutin, Christos E. Chadjichristos

**Affiliations:** 1INSERM, UMR-S1155, Bâtiment Recherche, Tenon Hospital, 75020 Paris, France; 2Faculty of Medicine, Sorbonne University, 75013 Paris, France; 3INSERM, UMR-942, MASCOT, Cardiovascular Markers in Stress Condition, Université de Paris, 75010 Paris, France; 4FHU PROMICE AP-HP, Saint Louis and DMU Parabol, Critical Care Medicine and Burn Unit, AP-HP, Department of Anesthesiology, Université Paris Cité, 75010 Paris, France

**Keywords:** kidney disease, experimental nephropathy, animal models, connexin 43

## Abstract

Renal disease is a major public health challenge since its prevalence has continuously increased over the last decades. At the end stage, extrarenal replacement therapy and transplantation remain the only treatments currently available. To understand how the disease progresses, further knowledge of its pathophysiology is needed. For this purpose, experimental models, using mainly rodents, have been developed to unravel the mechanisms involved in the initiation and progression of renal disease, as well as to identify potential targets for therapy. The gap junction protein connexin 43 has recently been identified as a novel player in the development of kidney disease. Its expression has been found to be altered in many types of human renal pathologies, as well as in different animal models, contributing to the activation of inflammatory and fibrotic processes that lead to renal damage. Furthermore, Cx43 genetic, pharmacogenetic, or pharmacological inhibition preserved renal function and structure. This review summarizes the existing advances on the role of this protein in renal diseases, based mainly on different in vivo animal models of acute and chronic renal diseases.

## 1. Introduction

Renal disease refers to a group of pathologies that affect the kidney’s structure and function. Acute and chronic kidney diseases are both common in adults and are associated with a high risk of morbidity and mortality [1].

Acute kidney injury (AKI) is characterized by a rapid decline in renal function. It occurs in about 10 to 15% of patients admitted to the hospital and affects 1 in 5 hospitalized patients worldwide [2]. AKI is more common in patients over 65 years old with co-morbidities such as cardiovascular disease, diabetes, liver disease, and pre-existing renal disease [1,3]. According to the KDIGO guidelines, chronic kidney disease (CKD) may occur after AKI and is defined by a decrease in the glomerular filtration rate of <60 mL/min/1.73 m^2^ for 3 months or more [1,4]. Some of the major risk factors for CKD include diabetes, high blood pressure, autoimmune disease, aging, and genetics [5]. CKD affects approximately 8 to 10% of the world population [5]. Patients with renal failure require individualized care based on their individual medical histories. Currently, AKI is treated according to the initial cause of the renal damage (i.e., sepsis, acute hemorrhage, hemodynamic shock, or inflammatory disease) and with a nephroprotection assessment, which promotes the rapid recovery of injured kidneys [6]. Concerning CKD, treatments rely mainly on limiting the disease progression and/or replacement therapies, such as extra-renal purification or renal transplantation at late stages [5,7].

Consequently, renal disease represents a real public health burden, with the current treatments, which are costly and sometimes not very effective, significantly altering the quality of life of patients. Although our understanding of the mechanisms involved in the development of kidney disease has improved in recent years, we are still searching for early prognostic markers and novel therapeutic options to prevent or reverse the progression of kidney disease. In this regard, the gap junction protein connexin 43 may be an attractive option.

## 2. Gap Junctions and Connexins

Cell–cell communication is essential for maintaining tissue homeostasis and structural integrity in multicellular organisms. Gap junctions are membrane channels that provide this type of communication in various organic systems, as they have been highly conserved through evolution [8]. These channels allow for the direct cytoplasmic exchange of ions and low-molecular weight metabolites (<1 kDa) between adjacent cells [9,10]. In vertebrates, this direct communication is mediated by the docking of two hemichannels, called connexons (Figure 1). The connexons are located at the plasma membrane of two adjacent cells and form a hydrophilic central pore that allows for the exchange of ions, including Na, K+, and Ca2+, small metabolites, and second messengers (cAMP, IP3, ADP, ATP, etc.) [8,11,12]. Uncoupled connexons also allow for communication between the cytoplasm and the extracellular environment [13]. Each connexon is composed of six homo- or hetero-hexameric connexin (Cx) subunits [11]. Connexins are a group of proteins named according to their approximate molecular weight (kDa). Extensive studies have identified 21 different isoforms in humans and 20 in mice. Most tissues and cells express these proteins, except for erythrocytes, sperm cells, and striated muscle cells [8]. Regarding their structure, Cxs cross the plasma membrane four times, producing two extracellular loops and one intracellular loop and exposing their N- and C-terminal tails in the cytosol [8]. This structure is highly conserved in the Cx family members. However, the intracellular loop and the N- and C-terminal tails have a large variation in their length and amino acids. Due to their structure and function in the intracellular exchange of metabolites and other second messengers, gap junctions participate in many physiological processes, such as cell proliferation, growth, or differentiation [14,15]. It is also evident that they are involved in angiogenesis, glandular secretion, and neuronal migration [16,17,18].

In recent years, several studies have suggested that Cxs are involved in renal pathophysiology. Despite being well documented, these proteins are still a challenge, both in terms of their involvement as novel therapeutic targets and in their pathophysiological mechanisms. Previous studies have shown that certain isoforms are expressed in the kidneys, including Cx26, Cx30.3, Cx31, Cx32, Cx37, Cx40, Cx43, Cx45, and Cx46 [19,20]. Many studies have focused on their function and localization, but these data are still controversial. In this review, we focus mainly on the role of Cx43 in renal disease.

## 3. Connexin 43

Almost ubiquitously expressed, Cx43 is the most documented constitutive gap junction protein. The gene encoding for Cx43 is the gap junction alpha-1 protein (GJA1) [21]. Its C-terminal part differs from other Cxs by the presence of about twenty phosphorylation sites and protein interaction domains, allowing it to play different physiological roles; in addition to its role in cell–cell communication, Cx43 can mediate gene transcription, cytoskeleton dynamics, ATP exocytosis, vesicle release, and cellular stress (Figure 1) [13,22]. These functions are due to the unique structure of Cx43, which is composed of an untranslated exon 1 separated by an intron from exon 2, containing an uninterrupted coding region and a 3′ untranslated region (3′-UTR) [21]. Moreover, Cx43 is abundantly expressed in ventricular cardiomyocytes and, more precisely, in intercalated discs, allowing it to electronically couple adjacent myocytes and synchronize the cardiac action potential [23,24]. Impaired trafficking can lead to arrhythmias and sudden cardiac death [25]. Many authors have investigated its role in different types of heart disease, notably atherosclerosis, where an increase in Cx43 expression has been found with a modification of its localization [26]. However, an inhibition of its expression in mice susceptible to atherosclerosis was protective, probably by limiting inflammation [22]. Other cell types express Cx43, notably astrocytes. Studies on human post mortem brains suffering from Alzheimer’s disease, and equivalent models in mice, have also shown an increased expression of Cx43 leading to neurotoxic effects through the excessive release of ATP and glutamate [27]. Studies on experimental animal models have also shown a role for Cx43 in retinal diseases. Indeed, a high expression of this Cx has been found in a diseased human eye (diabetic retinopathy, epithelioid and mixoid melanoma, etc.), playing a role in the establishment of the inflammatory process and the proliferation of tumor cells [28].

The implication of Cx43 in the inflammatory process has also been demonstrated in acute lung injury, contributing to pulmonary oedema [29]. In view of its involvement in a variety of pathological processes, the inhibition of Cx43 has shown promising results. Indeed, the use of Gap27, a pharmacological inhibitor of Cx43, in a mouse model of Parkinson’s disease was able to reduce the damage of dopaminergic neurons [30]. In addition, pharmacogenetic inhibition using antisense oligonucleotides limited inflammation and promoted wound healing [31]. While previous studies using Cx43 channel blockers reported improved outcomes, some studies in knockout animals showed the opposite results. Indeed, in a diabetic mouse model, a decrease in Cx43 expression was associated with the development and progression of diabetic retinopathy [32]. Finally, Cx43 is expressed in various cancers, including colorectal, breast, and cervical cancers [14,33,34]. However, the current studies remain controversial, showing that its expression is both protective and pro-metastatic, depending on the type of tumor [35].

## 4. Connexin 43 in the Kidney

The research on connexins has been an important field of biomedical research since their discovery in the mid-1970s, but Beyer et al. were the first to demonstrate Cx43 expression in rat kidneys in 1989 [36]. Later studies determined the localization of this protein in mouse models in the renal vasculature (endothelial cells), the glomerulus (podocytes and mesangial cells), and in the collecting duct [37,38]. Numerous studies have since focused on the role and molecular mechanisms mediated by Cx43 in different renal compartments where it is expressed and on its function in renal injury. This review focuses on experimental murine models of AKI and CKD that have helped us understand the role of this Cx in these diseases.

### 4.1. Connexin 43 in Experimental AKI

A few studies have been performed on biopsies from patients showing a variation in Cx43 expression in the kidneys [39,40]. Since then, it has become important to understand the role of this Cx in the functional and structural adaptations of the kidney in response to acute injury. The rodent models of AKI, which were developed for this purpose, will be discussed.

#### 4.1.1. Uranyl Acetate

Hishida et al. were the first to explore the expression of Cx43 in an animal model of AKI induced by a uranyl acetate (UA) injection. UA is a derivative of uranium, a radioactive heavy metal whose acute exposures are chemically toxic to the kidneys, causing various reversible disorders, mainly characterized by acute damage to the proximal tubule [41].

To determine the mechanisms of differentiation of myofibroblasts from interstitial fibroblastic cells during AKI, Hishida et al. induced acute renal failure in Sprague–Dawley rats by intravenous injection of UA and explored its consequences 2 and 21 days after the injection. In healthy kidneys, rare immunofluorescent spots for Cx43 in the interstitium were observed. However, from day 2 onwards, more extensive staining was seen in the peritubular region, which is the main affected area following the induction of AKI by UA. The authors speculated that the upregulation of Cx43 could promote AKI by participating in the communication between myofibroblasts located around the damaged proximal tubules. However, since UA is radioactive and due to increasing safety concerns and restrictions imposed by government agencies, many authors have focused on more suitable alternatives [42].

#### 4.1.2. Lipopolysaccharide

Lipopolysaccharide (LPS) is an endotoxin derived from the outer leaflet of the outer membrane of Gram-negative bacteria, such as *Escherichia coli*, that has been used by many researchers to produce experimental pathology in animals [43,44]. In the murine kidney, the injection of LPS results in an elevation in plasma urea and creatinine, as well as an increase in certain inflammatory cytokines, reversibly modifying renal function [45,46].

De Maio et al. were the first to describe a model using this endotoxin in the kidney to study Cx43 expression in adult male rats. The LPS injection induced rapid and reversible acute inflammation, which allowed for the observation of the increase in Cx43 expression only 2 h after the injection; it reached a maximum at 4 h before returning to a basal state after 12 h. The authors suggested that the Cx43-increased expression was part of an inflammatory response induced by the LPS administration [47]. Qin et al. then studied the evolution of Cx43 expression, but also the effect of its pharmacological inhibition after an LPS injection. The study was performed in adult mice to determine if Fangjifuling (FF), a traditional anti-inflammatory medicinal plant used in China, could limit the AKI induced by LPS. Although the Cx43 expression was insignificant in the renal cortex of the control mice, it was abnormally elevated around the proximal and distal tubules after the LPS injection, as previously reported by Hishida et al. In addition, its inhibition promoted the nephroprotective effect of FF against LPS-induced inflammatory and apoptotic responses; thus, confirming the involvement of Cx43 in inflammatory and apoptotic processes [48].

The above-mentioned studies were mainly interested in the expression and localization of Cx43 during an LPS-induced AKI. This model was later used to understand the role of Cx43 in inflammatory responses via inflammasome activation by Yao et al. The activation of the NLRP3 inflammasome is a major cellular event contributing to the production of inflammatory mediators. In this study, adult WT (Cx43+/+) and heterozygous (Cx43+/−) mice, in which the connexin 43 expression was genetically reduced by half, were injected intraperitoneally with LPS at different time points. As expected, the LPS administration induced renal inflammation and tubular damage, as well as it increased the levels of IL-1B, blood urea nitrogen (BUN), and proteinuria. This renal dysfunction was associated with an increased expression of Cx43 and an increase in the production of reactive oxygen species (ROS). Thus, the authors suggested that Cx43 contributes to the activation of the inflammasome through the modulation of the intracellular redox status. This process was regulated by Cx43 via Ca2+ signaling, a crucial molecule involved in ROS induction, but also via ROS propagation between adjacent cells. Thus, using this animal model allowed for the partial explanation of the role of Cx43 in inflammatory responses and inflammasome regulation [49].

#### 4.1.3. Cisplatin

Cis-diamminedichloroplatinum (II) (cisplatin) is a drug used to treat several types of cancer. This chemotherapeutic agent contains platinum and forms complexes when it binds to DNA, causing the DNA strands to cross-link, leading to programmed cell death. Although it is an effective treatment for slowing down the growth of cancers, it is known to cause nephrotoxicity. Indeed, cisplatin-induced AKI in mice mimics manifestations of clinical AKI, such as decreased renal function and increased expression of tubular injury biomarkers [50]. Based on these observations, this model has been used to improve our knowledge of the physiology and pathophysiology of Cx43 in AKI.

Yu et al. focused on the regulatory effect of Cx43 on cisplatin-induced ferroptosis in mice. Indeed, ferroptosis is one of the main causes of cisplatin-induced AKI. Basically, it results from excessive iron-dependent lipid peroxidation that induces cell death. The expression level of Cx43 was significantly increased in cisplatin-injected mice, followed by an increase in tubular injury biomarkers, such as neutrophil gelatinase-associated lipocalin (N-GAL), kidney injury molecule-1 (KIM-1), and plasma urea [51]. Interestingly, the pharmacological inhibition of Cx43 by the Gap27 blocking peptide reduced the Cx43 levels and significantly decreased the ferroptosis markers. Moreover, in this study, human kidney proximal tubular cells (HK2 cells) were used and the Cx43 expression increased in a dose-dependent manner, while its downregulation decreased apoptosis in these cells. In addition, the Cx43 decrease resulted in an increase in SLC7A11 and GSH, which are involved in the inhibition of ferroptosis. In conclusion, the downregulation of Cx43 expression reduced AKI in this mouse model by inhibiting cisplatin-induced ferroptosis. However, further research into this mechanism is needed to provide more information on the treatment of AKI [52].

### 4.2. Connexin 43 in Experimental CKD

Previous studies have shown that Cx43 is expressed in the renal parenchyma and vasculature under resting and/or pathological conditions in animals and humans [36,39]. The importance of Cx43 in cell signaling and communication led to its investigation in renal pathology, particularly CKD. Regardless of the initial damage type, glomerular, endothelial, or tubulointerstitial, the main features of CKD are the recruitment of inflammatory cells and renal fibrosis [53]. Consequently, this review focuses on the role of Cx43 in these major CKD effects in different models of experimental nephropathy.

#### 4.2.1. Cx43 and Glomerular Damage

The role of Cx43 using models of experimental glomerulopathies has been reported mainly in rodents and targeted cells, particularly podocytes, specialized in the maintenance of the glomerular filtration barrier. Glomerulonephritis (GN) represents a variety of pathologies affecting the kidney and, more precisely, the glomerulus. It is the second most common cause of CKD, often progressing to end-stage renal failure. Several causes are at the origin of this pathology, such as infections, genetics, and hereditary or autoimmune diseases. In addition, some studies reported various information, which allowed for the highlighting of glomerular lesions that can subsequently affect other renal compartments [54].

There are different types of GN; the most severe and rapidly progressive being characterized by crescent formation in the human kidney, affecting less than 50% of the total glomeruli [55]. The progression of GN is characterized by detachment, followed by loss and/or migration of podocytes to the urinary chamber; however, the underlying molecular and cellular mechanisms remain unclear.

##### Induction of Glomerulonephritis by Injection of Nephrotoxic Serum

In our group, GN is induced by the injection of a nephrotoxic serum (NTS), according to a protocol popularized by Salant [56]. The NTS can cause crescents and renal fibrosis, along with proteinuria and glomerular damage. We have demonstrated an abnormal expression of Cx43 in injured glomeruli in mice after an NTS injection [57]. Interestingly, Cx43 expression was mainly detected in the glomerular endothelial tuft and expressed de novo in podocytes. This increase promoted glomerular damage, contributing to disease progression. Furthermore, the inhibition of this protein using Cx43+/− mice, or the specific blockade with Cx43 antisense, was able to preserve the structure and function of the kidney. Finally, we proposed a mechanism by which proinflammatory and/or profibrotic cytokines activated the expression of Cx43 during the progression of the disease at the transcriptional level in podocytes, leading to de novo expression of Cx43 hemichannels, contributing to an exacerbated ATP release. ATP, in turn, could mediate via purinergic signaling podocyte damage, illustrated by either foot process effacement or cell death [58].

##### Induction of Glomerulonephritis by Anti-Thymocyte Serum (ATS) Injection

Morioka et al. were interested in another type of GN induced by an anti-thymocyte serum (ATS) injection. Mesangioproliferative GN was induced in rats by the injection of antibodies against the Thy1-1 antigen, present only on the surface of the mesangial cells of the rats, thus, allowing the establishment of mesangiolysis, followed by an expansion of the mesangial extracellular matrix. Mesangial cells and podocytes expressing Cx43 showed no change in its expression during the progression of this GN model [59].

##### Puromycin Aminonucleoside

Puromycin aminonucleoside (PAN) is a widely used model of nephrotic syndrome progressing to focal segmental glomerulosclerosis. PAN is known to cause a nephrotic syndrome characterized by glomerular morphological changes and severe proteinuria, but the extent of damage varies with the dose of PAN administration [60,61].

Previous studies performed by Yamamoto et al. demonstrated an increased Cx43 expression along the glomerular capillary wall in rats at early stages of PAN-induced disease. Further studies using this model confirmed that the increased expression of Cx43 was one of the earliest responses to podocyte injury ever observed. Furthermore, microinjections of the fluorescent dye Lucifer yellow in cultured podocytes suggested that Cx43-mediated gap junctional intercellular communication was present between podocytes. The authors proposed that these cells may behave as an integrated epithelium within the glomerulus, rather than individually as separate cells in response to injury [62].

##### Phosphate Overload-Induced Glomerulonephritis

Sekiguchi et al. studied Cx43 expression in a mouse model of phosphate overload leading to GN. Indeed, transgenic rats with overexpression of Pit-1, a type III Na-dependent phosphate transporter, were used to establish this animal model as a model for the study of podocyte damage. The uptake of phosphate at the cell membrane is critical for cell viability, but a phosphate overload may also damage the cells. Consequently, this model resulted in severe proteinuria due to the phosphate-dependent podocyte injury and damage to the glomerular barrier, leading to the development of glomerulosclerosis. An interesting finding in this model was the appearance of enhanced immunofluorescence for Cx43 in podocytes prior to any renal dysfunction being observed. Thus, the authors considered Cx43 increased expression as an early signal of podocyte damage in transgenic rats [63].

##### Podocyte Injury via Aldosterone Injection

Aldosterone is originally a mineralocorticoid hormone secreted in the zona glomerulosa of the adrenal cortex in response to angiotensin-2 stimulation or elevated blood potassium and is associated with the development of podocyte injury through oxidative stress induction. This is associated with albuminuria, hypertension, and the development of glomerulosclerosis. Based on these observations, Yang et al. performed aldosterone injections in mice to study the effects of Cx43 on podocyte injury and explore the possible molecular mechanism behind this effect. They demonstrated that the upregulation of Cx43 was mediated by aldosterone, which could facilitate the propagation of apoptotic signals with an increase in ROS production and the Bax/Bcl-2 ratio [64].

##### High-Fat Diet

Obesity has been widely recognized for several years as a result of kidney damage. This disease is characterized by an excess of body fat gained from excessive dietary energy intake. Obesity contributes to the increasing prevalence of glomerulopathy and accelerates the progression of CKD. Altered fatty acid and cholesterol metabolisms are emerging as key mediators, contributing to this pathology with renal lipid accumulation, inflammation, oxidative stress, and fibrosis.

In a study performed by Zhao et al., the expression of Cx43 was investigated in a high-fat diet model in rats for a long period of time. The hypercaloric intake caused glomerular hypertrophy and podocyte foot effacement, as well as an elevation in the podocyte injury markers. The Cx43 expression was also found to be increased within the injured glomeruli and was correlated with obesity-related inflammation, suggesting that this Cx may be a potential target in the development of obesity-related glomerulopathy [65].

#### 4.2.2. Cx43 in Obstructive Nephropathy: The Unilateral Ureteral Obstruction Model

The unilateral ureteral obstruction model (UUO) has been currently used in rodents. The UUO consists of a ligation of the left ureter with the unobstructed right kidney serving as a control. This model presents fundamental pathogenic mechanisms that characterize some forms of CKD, in addition to tubular injuries resulting from obstructed urine flow, allowing, in the long-term, for severe damage to the renal structure and further renal fibrosis.

Sommer et al. first observed that Cx43 mRNA expression was strongly increased in ureteral ligated kidneys in rats 5 days after the UUO, indicating a possible function of this protein in the remodeling of the renal tissue after tubular damage and fibrosis [66].

By using the same model, our group confirmed an increased Cx43 expression in the early stages of renal disease in mice. Moreover, a decrease in its expression via the use of Cx43-specific antisense or in Cx43+/− mice considerably limited the recruitment of inflammatory cells and cell adhesion molecules. Regarding the fibrotic process, which occurs in response to inflammation, the decreased Cx43 expression limited the extracellular matrix remodeling, probably by inhibiting the activation of the TGFB1/ERK pathway. Consequently, Cx43 seems to play a crucial role in the inflammatory process, as also reported in AKI [48,49], but also in the progression of renal fibrosis [67]. More recently, Price et al. showed that the increased expression of Cx43 after an UUO was accompanied by a reduction in the expression of E-cadherin and ZO-1, markers commonly associated with epithelial phenotypes, and an increase in N-cadherin and β-catenin, mainly associated with mesenchymal phenotypes, reflecting an epithelial–mesenchymal transdifferentiation. This transition was blunted in Cx43+/− mice. Moreover, an augmentation of inflammatory markers, such as NLRP3, was also reported in the same mice, as well as an increase in the ATP release mediated by Cx43 hemichannels, which may promote tubular lesions and the establishment of the inflammatory process [68]. Indeed, additional studies have shown that the release of ATP activated purinergic receptors, such as P2 × 7, which contribute to fibrosis in several types of diseases [69].

#### 4.2.3. Cx43 and Hypertensive Nephropathy

High blood pressure is one of the main causes of CKD. This hypertension can be primary or secondary to the development of the disease. It is widely recognized that the kidney plays a crucial role in the regulation of blood pressure. In this section, we report the main hypertensive animal models used in studies to determine Cx43 functions in this pathological condition.

##### Stimulation of Renin Secretion

Haefliger et al. used a hypertensive rat model to study the distribution of Cx43 in the kidney by clipping the renal artery to stimulate renin secretion. In this context, Cx43 was expressed in renal vessel endothelial cells at moderate levels [70]. Following this observation and the potential expression of Cx43 in endothelial cells, Garcia-Pinto et al. also used this model to study its distribution more precisely. A Cx43 expression was observed in the interdigitations and in the cytoplasm of endothelial and tubular cells. However, in spontaneously hypertensive rats, Cx43 expression was found to be decreased [71]. Over time, several authors showed contradictory data. To study the role of Cx43 in hypertension-induced CKD, RenTg mice, a model of hypertension-induced CKD, where renin was expressed ectopically in the liver, were crossed with Cx43+/− mice. Interestingly, the upregulation of Cx43 contributed to structural damage in the renal cortex in the RenTg mice, and the RenTgCx43+/− mice showed improved tissue structure and function. Accordingly, Cx43 expression increased as renal disease progressed, whereas a Cx43 decrease in expression was beneficial in chronic inflammation [67].

##### Renal Injury via Angiotensin II

The renin–angiotensin system (RAS) is an endocrine network composed of renin and angiotensin II (AngII) that acts on the kidneys and adrenal glands, regulating blood pressure, intravascular volume, and electrolyte balance. Moreover, it is well established in the literature that AngII contributes to the rise in blood pressure, which causes progressive damage to nephrons, generating chronic lesions and further end-stage renal disease. The infusion of AngII in rats showed a possible correlation between the activation of the RhoA/ROCK pathway and the expression of Cx43, leading to an increase in pro-inflammatory cytokines, such as the tumor necrosis factor α (TNF-α) and interleukine-1β (IL-1β), and further kidney damage [72].

#### 4.2.4. Cx43 and Diabetic Nephropathy

Chronic hyperglycemia and high blood pressure are the main risk factors for the development of diabetic nephropathy (DN), which represents a major cause of CKD worldwide. There are many pathways and mediators involved in the development and progression of DN, but they still need to be further explored. Renal fibrosis is the main outcome of DN, which is characterized by glomerular sclerosis and tubulointerstitial fibrosis.

Different experimental models of DN have been used to study the expression and role of Cx43 in this pathology. Satriano et al. studied this protein in rats with type 1 diabetes induced by a streptozotocin (STZ) treatment, a cytotoxic substance on the beta cells of the islets of Langherans. They observed a decreased expression of Cx43 after the induction of DN [73]. Similar data were reported by Hu et al. and Zhang et al. [74,75]. The use of db/db mice with type 2 diabetes and leptin receptor deficiency also reported that elevated glucose levels reduced Cx43 expression. In addition, an increase in Cx43 expression improved the activation of the Nrf2/ARE pathway, a crucial cellular defense mechanism against oxidative stress, ultimately attenuating renal fibrosis in diabetic mice [76]. Additional studies performed in db/db mice also showed that Cx43 activation reduced the development of diabetic renal fibrosis by regulating EMT via the SIRT-1 HIF signaling pathway, involved in metabolism and inflammation. In the same study, overexpression of Cx43 inhibited EMT and reduced the expression of ECM components, such as fibronectin (FN), collagen I, and collagen IV, in NRK-52E (rat kidney proximal tubular epithelial cell line) cells induced by high glucose [77]. Finally, the use of boldine, a well-known anti-inflammatory and hypoglycemic alkaloid, in mesangial cells prevented the increase in oxidative stress, the decline in gap junction intercellular communication, and the increase in cell permeability due to Cx43 hemichannel activity resulting from high glucose and proinflammatory cytokines [78].

All of the above-mentioned observations on Cx43 expression and the proposed functions in different models of experimental nephropathy are listed in Table 1.

## 5. Connexin 43 and Human Nephropathy

The difficulty of obtaining samples reduces the number of human studies. Consequently, existing data are mainly based on the in vitro culture of human cells and a few renal biopsies of patients with different types of nephropathies.

Cx43 was first detected in human kidneys in 1997 by Hillis et al. The authors studied the expression of this protein in biopsies from patients with various inflammatory glomerulopathies and found increased expression of Cx43 in infiltrating and damaged tubular epithelial cells and de novo expression in interstitial injured sites [39]. The pattern of expression of Cx43 was closely paralleled by that of cell adhesion markers, and the authors speculated that this protein was implicated in tubulointerstitial inflammation.

Later, Sawai et al. reported that the altered expression of Cx43 was associated with progressive diabetic type 2 nephropathy in human biopsies. Similar data were reported in in vivo models [79]. Furthermore, Guo et al. found that diabetic patients showed decreased expression of Cx43 in the glomerulus when the MAPK/mTOR pathways were activated [80].

As previously shown in rodents, Kurtz et al. reported Cx43 expression in human kidneys, mainly in the endothelium of the arteries/arterioles and the juxtaglomerular apparatus [81]. Our research group reported that immunofluorescence for Cx43 was highly induced in biopsies of patients suffering from different types of nephropathies, such as IgA and C3 glomerulonephritis, nephroangiosclerosis, and obstructive nephritis [58,67].

Finally, a mechanism has been recently proposed based both on biopsies of CKD patients and in vivo and in vitro models of tubulopathy, underlying aberrant Cx43 hemichannel activity via ATP release, supporting purinergic receptors and inflammasome activation, thus, promoting the progression of the disease [82].

## 6. Conclusions

Chronic kidney disease is a major global health burden due to the increasing prevalence of its associated risk factors. Despite recent medical advances, end-stage renal disease remains a major challenge. Therefore, finding new therapeutic targets to detect renal disease earlier and improve patient outcomes is a medical priority. Among some candidates, the gap junction protein Cx43 has emerged as a promising therapeutic target. Indeed, even though its expression depends on the type of nephropathy, it appears crucial to the progression of inflammation and renal fibrosis. Furthermore, Cx43 inhibition, whether pharmacological, genetic, or pharmacogenetic, was able to limit renal damage and improve renal function in animal models of experimental nephropathy. Hence, new therapies targeting Cx43 could be developed to delay kidney disease progression in the near future.

## Figures and Tables

**Figure 1 ijms-23-13090-f001:**
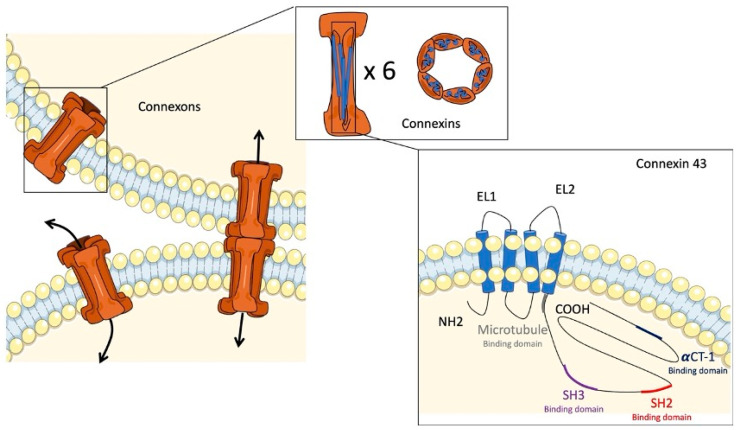
Gap junctions and connexin 43. Illustration of connexons binding together to form a gap junction channel with a central pore. The connexons are represented as cones composed of six connexin subunits. Connexin 43 is represented by four transmembrane domains, two extracellular loops (EL1 and EL2), one intracellular loop, and N-terminal (NH2) and C-terminal (COOH) tails.

**Table 1 ijms-23-13090-t001:** Physiopathological functions of Cx43 in different in vivo models of renal disease.

	Renal Compartment	Models	Effects
**AKI**	Proximal Tubules	Uranyl Acetate	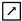 Cx43 expression
Tubules	Lipopolysaccharide	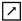 Cx43 expression 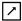 Tubular damage, inflammation, BUN and proteinuria 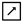 NLRP3 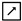 ROS
Cisplatin	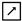 Cx43 expression 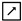 Tubular injury
**CKD**	Glomerular Compartment	Nephrotoxic Serum (NTS)	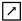 Cx43 expression 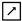 Proteinuria and glomerular damage (crescents) 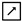 Renal fibrosis 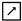 Proinflammatory and/or profibrotic cytokines
Anti-thymocyte Serum (ATS)	No change in Cx 43 expression
Puromycin aminonucleoside	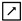 Cx43 expression 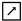 Podocytes injury
Phosphate Overload	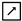 Severe proteinuria 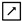 Cx43 expression in podocytes
Aldosterone injection	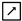 Cx43 expression in podocytes 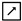 ROS production and Bax/Bcl-2 ratio
High Fat Diet	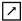 Cx43 expression 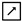 Inflammation
Tubulo-interstial Compartment	Unilateral Ureteral Obstruction (UUO)	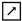 Cx43 expression 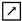 Renal fibrosis and inflammation (NLRP3) 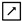 ATP release 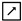 N-cadherin and β-catenin 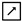 E-cadherin and ZO-1
Vascular Compartment	Stimulation of renin secretion	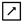 Cx43 expression 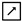 Structural damage and chronic inflammation
Stimulation of Angiotensin II	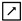 Cx43 expression 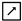 Proinflammatory cytokines 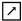 RhoA/ROCK pathway
Diabetic Mice	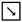 Glucose levels 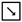 Cx43 expression

## Data Availability

Not applicable.

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
