# Peer review of "The Role of Connexin 43 in Renal Disease: Insights from In Vivo Models of Experimental Nephropathy"

_ijms, 2022, doi:10.3390/ijms232113090_

Round 1

Reviewer 1 Report

This very well-organized review gathers most if not all relevant contributions on topics described in the manuscript.

I just like to suggest that in the section related to diabetes the following manuscript could be included One of the earliest demonstrations of importance of Cx hemichannels in kidney alteration in diabetes was published in doi: 10.1155/2013/593672

Author Response

We thank the reviewer for his/her kind remarks. To address its concern we have mentioned the publication of Hernández-Salinas et al and added as reference 78 in the revised version of our manuscript: 

[78] Hernández-Salinas, A. Z. Vielma, M. N. Arismendi, M. P. Boric, J. C. Sáez, et V. Velarde, « Boldine Prevents Renal Alterations in Diabetic Rats », J. Diabetes Res., vol. 2013, p. 593672, 2013, doi: 10.1155/2013/593672.

All modifications are indicated in red.

Reviewer 2 Report

The authors reviewed a lot of previous study regarding the influence of Connexin 43 (Cx43) on renal disorder. They explained the role and physiological effects of Cx43 on the development of renal disease ranged from acute kidney injury (AKI) to chronic kidney disease (CKD). In addition, they introduced several previous papers in which influence of Cx43 on renal dysfunction was evaluated by using in vivo model of AKI or CKD such as CP-N or DKD model.

Manuscript was well written and clearly explained. I could understand the value of Cx43 as biomarker of activity and severity of renal disease or therapeutic potential of Cx43 in the treatment of renal disease. I found no specific contradiction.

Author Response

We thanks the reviewer for his/her encouraging comments.

 In the revised manuscript, modifications are indicated in red.
